# UPRISE: Universal Prompt Retrieval for Improving Zero-Shot Evaluation

**Daixuan Cheng   Shaohan Huang*  Junyu Bi   Yuefeng Zhan   Jianfeng Liu**
**Yujing Wang   Hao Sun   Furu Wei   Denvy Deng   Qi Zhang**
Microsoft Corporation

daixuancheng6@gmail.com     bijunyu21@mails.ucas.ac.cn

{shaohanh, yuefzh, jianfengliu, yujing.wang, hasun, fuwei, dedeng, qizhang}@microsoft.com

## Abstract

Large Language Models (LLMs) are popular for their impressive abilities, but the need for model-specific fine-tuning or task-specific prompt engineering can hinder their generalization. We propose UPRISE (**U**niversal **P**rompt **R**etrieval for **I**mproving zero-**S**hot **E**valuation), which tunes a lightweight and versatile retriever that automatically retrieves prompts for a given zero-shot task input. Specifically, we demonstrate universality in a cross-task and cross-model scenario: the retriever is tuned on diverse tasks, but tested on unseen task types; we use a small frozen LLM, `GPT-Neo-2.7B`, for tuning the retriever, but test the retriever on different LLMs of much larger scales, such as `BLOOM-7.1B`, `OPT-66B` and `GPT3-175B`. Additionally, we show that UPRISE mitigates the hallucination problem in our experiments with `ChatGPT`, suggesting its potential to improve even the strongest LLMs. Our model and code are available at https://github.com/microsoft/LMOps.

## 1 Introduction

Large Language Models (LLMs) such as `GPT-3` (Brown et al., 2020), `OPT` (Zhang et al., 2022), and `BLOOM` (Scao et al., 2022) have shown impressive capabilities across a wide range of tasks. Recent research proposes two main approaches to further improve their performance: fine-tuning LLMs to follow prompts (Hu et al., 2022; Houlsby et al., 2019; Zaken et al., 2022; Wei et al., 2022a; Sanh et al., 2022) and developing prompt engineering techniques to guide the LLMs (Brown et al., 2020; Wei et al., 2022b; Liu et al., 2021; Lester et al., 2021).

Fine-tuning LLMs adjusts their weights to fit specific prompts. However, this can be constrained by computational limitations or the unavailability of model weights (Hu et al., 2022). Multi-task tuning

---
*Corresponding author

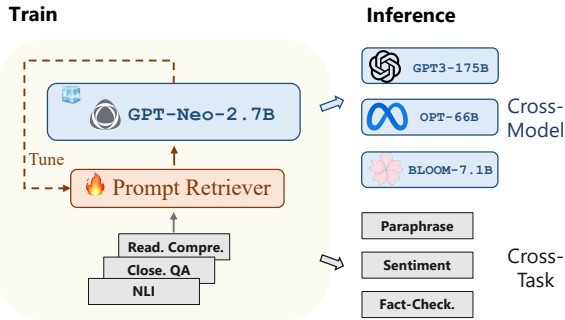

Figure 1: UPRISE tunes a prompt retriever on multiple tasks with a small LLM, but conducts inference on unseen task types with a different larger LLM.

provides an alternative approach to improve zero-shot task generalization (Wei et al., 2022a; Sanh et al., 2022), which partially justifies the tuning cost. Yet, the constant evolution of LLMs creates the need for tuning new models, making the cumulative fine-tuning cost a big concern.

Prompt engineering constructs prompts to guide frozen LLMs. Prompt design adds an engineered natural language prompt to teach the LLM to learn in context (Brown et al., 2020) or to induce the LLM to reason (Wei et al., 2022b). Prompt tuning adds a soft prompt represented by continuous parameters, and optimizes it through gradient propagation (Liu et al., 2021; Li and Liang, 2021; Lester et al., 2021). While these methods can improve performance for specific tasks, it is uncertain whether the prompts designed for one task can generalize to unseen task types, as prompt designers are blind in strict zero-shot settings (van de Kar et al., 2022).

In this paper, we propose UPRISE (**U**niversal **P**rompt **R**etrieval for **I**mproving Zero-**S**hot **E**valuation), which tunes a lightweight and versatile retriever that automatically retrieves prompts from a pre-constructed pool, given a zero-shot task input. As illustrated in Figure 1, the retriever is trained to retrieve prompts for multiple tasks, enabling it to generalize to un-

seen task types during inference. In addition, we demonstrate that the cross-task capabilities can generalize well from a small LLM to different LLMs of much larger scales: we use `GPT-Neo-2.7B` (Black et al., 2021) to guide the tuning of the retriever and evaluate the retriever's performance on `BLOOM-7.1B` (Scao et al., 2022), `OPT-66B` (Zhang et al., 2022), and `GPT3-175B` (Brown et al., 2020). The cross-model and cross-task generalization of UPRISE makes it a promising and practical solution for real-world applications.

Furthermore, our approach demonstrates the potential for enhancing even the most powerful LLMs, as shown in our experiments with `ChatGPT`. Despite its impressive abilities, `ChatGPT` has been found to struggle with serious hallucination problems, leading to responses that are factually inaccurate (Bang et al., 2023). However, UPRISE is able to address this issue on fact-checking tasks by prompting the model to draw correct inferences from its built-in knowledge.

In summary, our contributions include:

- We introduce UPRISE, a lightweight and versatile approach to improve zero-shot performance of LLMs in the cross-task and cross-model scenario.
- UPRISE is tuned with `GPT-Neo-2.7B`, but can also benefit different LLMs of much larger scales, such as `BLOOM-7.1B`, `OPT-66B`, and `GPT3-175B`.
- Our exploration on `ChatGPT` demonstrates the potential of UPRISE in improving performances of even the strongest LLMs.

## 2  Problem Definition

We aim to improve zero-shot performance of LLMs by training a prompt retriever to retrieve prompts[1] for any given task input. Specifically, UPRISE decomposes the prompting process into two steps: *retrieve* then *predict*. Given an input $x$, we first retrieve a set of positive prompts $\mathcal{P}^+$ from a pre-constructed pool $\mathcal{P}$:

$$\mathcal{P}^+ = \mathcal{R}(x, \mathcal{P}). \qquad (1)$$

Then we concatenate $\mathcal{P}^+$ with $x$ to form an input sequence for a frozen LLM, which generates a predicted output:

$$y^{\mathcal{P}^+} = \mathrm{LM}\left(y^{\mathcal{P}^+} | \mathcal{P}^+ \oplus x\right). \qquad (2)$$

[1] "Prompt" sometimes refers to a natural language template filled by an input example, but here it denotes the sequence prepended to the task input.

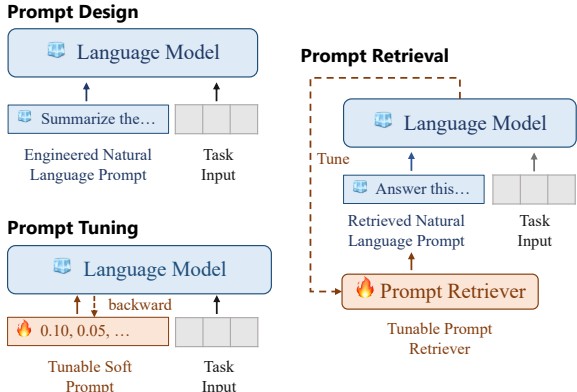

Figure 2: Typical prompt engineering methods and prompt retrieval. Prompt retrieval prepends a natural language prompt to the task input and uses a frozen LLM to evaluate the prompt's performance. The obtained evaluation is then used to tune the retriever in a reverse manner.

Our objective is to optimize performance of $y^{\mathcal{P}^+}$ to match the target $y$ by updating the retriever $\mathcal{R}$.

Figure 2 compares prompt retrieval with typical prompt engineering methods: prompt design adds an engineered natural language prompt (Brown et al., 2020; Wei et al., 2022b) and prompt tuning tunes a soft prompt (Liu et al., 2021; Lester et al., 2021). In contrast, prompt retrieval tunes a retriever to retrieve natural language prompts, which is both interpretable and flexible. It uses the language model itself to label each prompt in the pool as positive/negative, and then tunes a retriever from this signal (Rubin et al., 2022). Such fine-tuned prompt retrieval has demonstrated effectiveness in the task-specific scenario (Rubin et al., 2022; Ye et al., 2023): a prompt retriever is tuned on one or multiple specific tasks using the training sets as the prompt pool. The retriever is then evaluated on the corresponding testing sets.

Our work is to achieve universality of the prompt retriever, which means the fine-tuned retriever can be directly used to retrieve prompts for unseen tasks and various inference LLMs, without the need for further tuning. We define the universality from two perspectives: cross-task retrieval and cross-model retrieval.

**Cross-task retrieval.** Considering the diversity of tasks in real-world applications, we propose cross-task retrieval to retrieve for task types on which the prompt retriever has not been trained. We simulate this setting by evaluating the prompt retriever on unseen task types: various tasks are

grouped into different clusters based on their task types, and we hold out each task cluster for evaluation while training the retriever on all remaining clusters (Wei et al., 2022a).

**Cross-model retrieval.** Due to the high cost of tuning a prompt retriever with a large-scale LLM, we propose evaluating the capability to generalize from a small LLM to a large LLM. Specifically, we use a relatively small LLM for tuning the retriever, while using a much larger LLM for inference. Furthermore, we suggest exploring the transferability between different LLM sources, as there are LLMs developed by different companies or institutions.

## 3 Method

As shown in Figure 3, UPRISE uses a frozen LLM to supervise the fine-tuning of a prompt retriever on diverse tasks, and then uses this trained retriever to retrieve prompts for unseen task types with different LLMs during inference. In this section, we elaborate on our data construction, prompt scoring, retriever tuning and inference pipeline.

### 3.1 Data Construction

**Task Data.** We use instruction templates from FLAN (Wei et al., 2022a) to convert task datasets into natural language instructions[2]. Each task dataset corresponds to approximately seven templates. For each data example $(x_i, y_i)$, we randomly select one of the seven templates to convert $x_i$ into a task input and $y_i$ into a label completion. The option suffices and new line characters "$\backslash n$" are automatically removed from the task input, to make the text format more similar to that of the pre-training corpus, improving prompting performance (van de Kar et al., 2022).

**Prompt pool.** For each testing cluster, the prompt pool used for retrieval is made up of training demonstrations of the remaining task clusters (i.e., the clusters for training the retriever). This is inspired by in-context learning (Brown et al., 2020), which presents a few training demonstrations before the task input to improve model performance. Each demonstration is a concatenation of the task input and the label completion. Our motivation is that the testing input may benefit from similar question types, topics, or reasoning chains in the retrieved demonstrations, despite that the testing

---

[2]We exclude templates that "turn the task around", such as asking a sentiment classification task to generate a movie review.

input and the demonstrations are of different task types.

### 3.2 Prompt Scoring

For each training example $(x_i, y_i)$ in the training clusters, we collect a set of positive and negative prompts from the prompt pool $\mathcal{P} = \{p_j\}_{j=1}^{N_\mathcal{P}}$, where the positive prompt indicates that the frozen LLM achieves good task scores conditioned on the prompt-input concatenation. We use these positive and negative labels to supervise the contrastive learning of the retriever.

We categorize all tasks into two question types: text completion and multiple choice (Brown et al., 2020), and use different methods to score the prompts for each training example.

**Text completion** is the question to do free-form completion. We calculate score of the prompt using the following equation:

$$\text{score}\left(p_j, x_i\right) = \text{metric}\left(y_i, y_i^{p_j}\right), \qquad (3)$$

where $y_i^{p_j} = \text{LM}\left(y_i^{p_j}|p_j \oplus x_i\right)$ is the model prediction based on the input concatenation $p_j \oplus x_i$, and $\oplus$ is a text delimiter "$\backslash n$". $\text{metric}\left(\cdot\right)$ is the function used to calculate the task metric score (e.g., F1 or ROUGE).

**Multiple choice** is the question to choose one correct completion from several options. Suppose there are $M$ options in a multiple choice question $\left(x_i, y_i, \{o_m\}_{m=1}^M\right)$, where $\{o_m\}_{m=1}^M$ is the option set and $o_{y_i}$ is the gold option. We feed the concatenation $p_j \oplus x_i$ to the LLM and calculate per-token likelihood of each option: $\text{LH}\left(o_m\right)$. The option with the highest likelihood is considered as the model prediction $y_i^{p_j}$ (Brown et al., 2020).

Accuracy of the prediction $\text{acc}\left(y_i, y_i^{p_j}\right)$ is a common metric for multiple-choice questions, but it only produces 0 or 1 for each example, making it hard to compare prompt effectiveness. To address this, we multiply the accuracy by the per-token likelihood of the gold option, which is normalized by the sum of the per-token likelihood of all options, to achieve a fine-grained comparison. The final score is formulated as:

$$\text{score}\left(p_j, x_i\right) = \text{acc}\left(y_i, y_i^{p_j}\right) \cdot \frac{\text{LH}\left(o_{y_i}\right)}{\sum_{m=1}^M \text{LH}\left(o_m\right)}. \tag{4}$$

**Prompt filtering.** Intuitively, to collect the positive and negative prompts for each training example, we need to score every prompt in the prompt

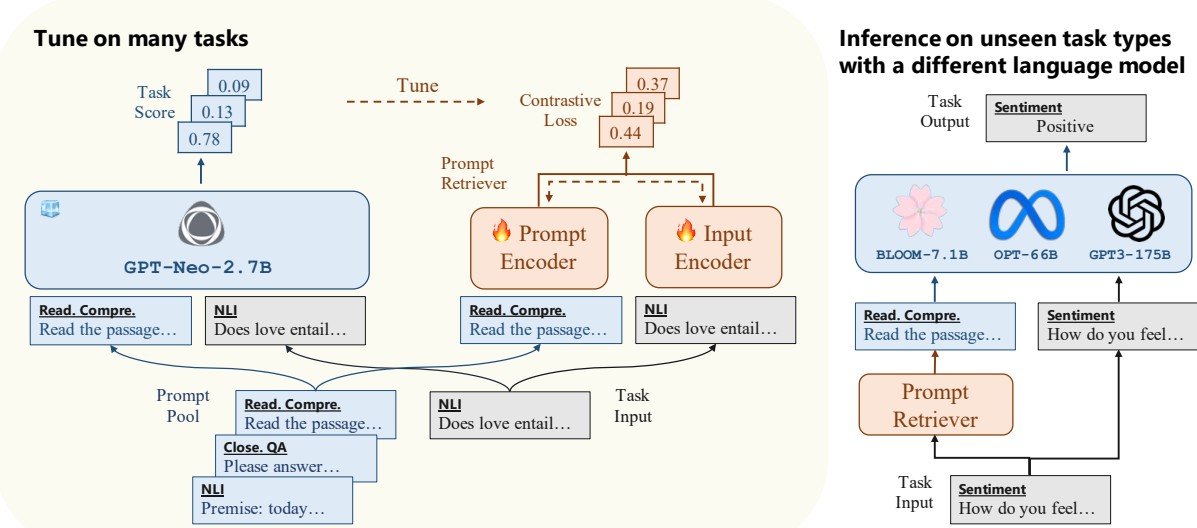

Figure 3: Training and inference pipeline. In the training stage, a frozen LLM is used to supervise the tuning of a prompt retriever, where both the LLM and the retriever take the prompt-input pairs as input, and we use the task scores given by the LLM to supervise the contrastive learning of the retriever. In the inference stage, for each task input, the tuned prompt retriever retrieve positive prompt(s) to guide the inference model to predict a task output. Overall, we follow a cross-task and cross-model paradigm where the task types and LLMs for training could be different from those for inference.

pool and identify the prompt that yields the best score as the positive prompt. Conversely, prompts that lead to the worst scores are labeled as negative prompts. However, scoring all the prompts can be computationally expensive (Rubin et al., 2022), even with a relatively small LLM.

To address this, we only score a subset of $L$ randomly sampled demonstrations; each demonstration is constrained to have the same task as the training example $(x_i, y_i)$. This is inspired by in-context learning where the testing sample and training demonstrations share the same task, resulting in improved task scores. By scoring a subset of demonstrations, we significantly reduce the computational cost while increasing the likelihood of identifying positive prompts within the sampled subset.

Furthermore, in the case of a difficult question, all $L$ prompt-input concatenation may result in a score of 0. To address this, we repeat the sampling process to score another subset of $L$ prompts with the same task as $(x_i, y_i)$, until we find at least one prompt with a score greater than 0.

For all the scored prompts for a training example, we label the prompt with the highest score as positive. For negative samples, we randomly sample $B$ training demonstrations from the prompt pool, each with a different task from that of $(x_i, y_i)$. In

addition, we label $B$ demonstrations corresponding to the lowest $B$ scores in the sampled prompts as hard negatives, which are of the same task with $(x_i, y_i)$ but are less effective.

### 3.3 Retriever Tuning

After labeling prompts for each training example, we split the collected data into two sets: $90\%$ for training and $10\%$ for validation. The prompt retriever is a bi-encoder model (Karpukhin et al., 2020) where the input encoder $E_X(\cdot)$ takes the task input $x_i$ as input, and the prompt encoder $E_P(\cdot)$ takes prompt $p_j$ as input.

To train the prompt retriever, InfoNCE (van den Oord et al., 2018) loss is used to maximize the similarity score between the encoded prompt and input for positive prompt-input pairs, and minimize it for (hard) negative prompt-input pairs. For a single training example $(x_i, y_i)$, the loss function for its positive and negative prompts is:

$$L(x_i, p_i^+, p_{i,1}^-, \ldots p_{i,2B}^-) \qquad (5)$$
$$= -\log \frac{e^{\text{sim}(x_i, p_i^+)}}{e^{\text{sim}(x_i, p_i^+)} + \sum_{j=1}^{2B} e^{\text{sim}(x_i, p_{i,j}^-)}},$$

where $p_i^+$ is the positive prompt, $p_{i,j}^-$ is one of the (hard) negative prompts, and $\text{sim}(x_i, p) = E_X(x_i)^\top E_P(p)$ calculates the similarity score be-

tween input $x_i$ and prompt $p$ using inner products (Rubin et al., 2022).

### 3.4 Inference

After fine-tuning the prompt encoder, we use it to encode the entire prompt pool with $E_P(\cdot)$. At inference time, for a testing task input $x_{\text{test}}$, we compute its encoding $E_X(x_{\text{test}})$ and then use maximum inner-product search over the prompt pool to retrieve $K$ most similar prompts, sorted by their inner product in descending order, denoted as $\mathcal{P}^+ = (p_1, ..., p_K)$. We then concatenate the prompts with the task input, resulting in the concatenation $p_K \oplus ... \oplus p_1 \oplus x_{\text{test}}$ (Rubin et al., 2022).

To evaluate the inference results, we use the same method described in Section 3.2 to generate predictions, and then use each task's corresponding evaluation metric to compute the scores.

## 4 Experiment Settings

**Task clustering.** We group the tasks used in our method into clusters, including Reading Comprehension, Closed-book QA, Paraphrase Detection, Natural Language Inference, Sentiment Analysis, Commonsense Reasoning, Coreference Resolution, Structure to Text, and Summarization. The datasets used in each cluster are listed in Appendix A.

**Data sampling.** To prevent the retriever tuning from being dominated by large datasets, we randomly sample up to $10k$ data examples from each task's training set, while also maintaining class balance in classification tasks[3]. The prompt pool consists of the sampled training data only. On average, for each testing task cluster, there are approximately $180k$ training examples sampled from the training clusters.

**LLMs.** We use `GPT-Neo-2.7B` (Black et al., 2021) from EleutherAI to tune the retriever, and evaluate the performance on larger LLMs from various sources during inference, including `BLOOM-7.1B` (Scao et al., 2022) from BigScience, `OPT-66B` (Zhang et al., 2022) from Meta, and `Davinci` and `text-davinci-001` from OpenAI, both belonging to the `GPT3-175B` (Brown et al., 2020) series. Greedy search is used to obtain predictions from all the LLMs.

**Prompt scoring.** We set the size of the randomly sampled subset to $L = 50$ and the number of (hard)

negatives to $B = 20$. For difficult questions, we repeat the re-sampling process up to seven rounds, as we found that this is sufficient to identify a positive prompt for 90% of the training examples. If no sampled prompt yields a score greater than 0, we filter out the corresponding training example.

[3]For instance, in a four-classification task, we sample a maximum of $2.5k$ data examples from each class.

**Tuning.** We initialize the two independent encoders of the retriever with $\text{BERT}_{\text{BASE}}$ (Devlin et al., 2019). Each retriever is fine-tuned for three epochs, and the best checkpoint is chosen based on retrieval accuracy using the validation set. For detailed tuning hyperparameters, Please refer to Appendix B.

**Inference.** During inference, we set the number $K$ of concatenated prompts to a relatively small value of 3, to balance between prompting performance and inference efficiency. For each dataset, we report metric scores on the test set when available, falling back to the validation set otherwise.

## 5 Main Results

We evaluate our prompt retriever on natural language understanding tasks where generative LLMs are known to need improvement (Liu et al., 2021). Table 1 compares the performance of UPRISE to vanilla zero-shot prompting.

### 5.1 Cross-Task Prompt Retrieval

Based on the results of `GPT-Neo-2.7B`, we can assess our ability of generalizing across different task types. UPRISE has positive impacts on most of the testing clusters. Specifically, we achieve absolute gains of 8.5% and 14.6% in Reading Comprehension and Paraphrase Detection tasks, respectively. We also find that UPRISE shows consistent performance improvements across all tasks in Closed-book QA and Natural Language Inference clusters.

However, UPRISE has negative impacts on Commonsense Reasoning and Coreference Resolution tasks. We conduct analyses in Appendix D to understand the reasons, revealing that Coreference Resolution hardly benefits from demonstrations and Commonsense Reasoning is harmed by different demonstration formats.

### 5.2 Cross-Model Prompt Retrieval

In addition to evaluating cross-task generalization, we can explore the cross-model ability by examining the results of `BLOOM`, `OPT`, `Davinci` and `text-davinci-001`. UPRISE continues to im-

| Task | Metric | GPT-Neo-2.7B | | BLOOM-7.1B | | OPT-66B | | Davinci | | Davinci-001 | |
|---|---|---|---|---|---|---|---|---|---|---|---|
| | | 0-SHOT | UPRISE | 0-SHOT | UPRISE | 0-SHOT | UPRISE | 0-SHOT | UPRISE | 0-SHOT | UPRISE |
| *Reading Comprehension* | | | | | | | | | | | |
| SQuADv1 | F1 | 4.4 | 26.4 | 4.5 | 5.5 | 6.1 | 7.5 | 6.5 | 6.0 | 41.6 | 57.7 |
| | EM | 0.4 | 14.3 | 0.0 | 0.0 | 0.0 | 0.6 | 0.0 | 0.0 | 16.4 | 36.8 |
| BoolQ | Acc | 54.5 | 59.4 | 54.0 | 60.2 | 60.7 | 63.5 | 62.0 | 65.7 | 64.2 | 65.7 |
| MultiRC | F1 | 57.1 | 58.1 | 58.8 | 59.8 | 59.6 | 60.4 | 59.8 | 60.0 | 54.3 | 58.9 |
| OBQA | Acc | 41.8 | 42.2 | 44.0 | 41.8 | 46.4 | 48.8 | 49.2 | 52.4 | 52.8 | 48.8 |
| **Average** | | 31.6 | **40.1** | 32.3 | **33.5** | 34.6 | **36.2** | 35.5 | **36.8** | 45.9 | **53.6** |
| *Closed-book QA* | | | | | | | | | | | |
| ARC-e | Acc | 45.7 | 55.6 | 53.7 | 60.9 | 56.2 | 66.0 | 64.1 | 71.8 | 67.0 | 74.4 |
| ARC-c | Acc | 29.3 | 30.0 | 33.2 | 34.2 | 36.7 | 40.2 | 40.8 | 45.2 | 46.2 | 50.4 |
| NQ | F1 | 1.3 | 5.6 | 0.9 | 1.4 | 2.5 | 2.1 | 0.0 | 2.2 | 18.3 | 18.2 |
| | EM | 0.5 | 2.2 | 0.0 | 0.1 | 0.3 | 0.4 | 0.0 | 0.0 | 4.8 | 8.7 |
| **Average** | | 19.2 | **23.3** | 22.0 | **24.2** | 23.9 | **27.2** | 26.2 | **29.8** | 34.1 | **37.9** |
| *Paraphrase Detection* | | | | | | | | | | | |
| MRPC | Acc | 46.6 | 67.9 | 51.0 | 70.6 | 51.0 | 68.9 | 54.4 | 62.3 | 40.0 | 61.3 |
| | F1 | 46.0 | 80.4 | 58.0 | 82.1 | 57.8 | 81.5 | 68.9 | 81.4 | 39.2 | 72.9 |
| QQP | Acc | 48.4 | 54.3 | 49.5 | 53.1 | 50.5 | 49.7 | 55.2 | 52.4 | 60.9 | 62.6 |
| | F1 | 42.2 | 59.8 | 46.7 | 59.6 | 43.7 | 58.5 | 33.7 | 57.9 | 43.0 | 45.9 |
| PAWS | Acc | 51.7 | 45.7 | 50.8 | 45.9 | 50.5 | 44.4 | 52.4 | 44.5 | 53.2 | 52.3 |
| **Average** | | 47.0 | **61.6** | 51.2 | **62.3** | 50.7 | **60.6** | 52.9 | **59.7** | 47.3 | **59.0** |
| *Natural Language Inference* | | | | | | | | | | | |
| MNLI-m | Acc | 35.3 | 41.3 | 35.4 | 36.0 | 37.0 | 40.4 | 34.2 | 38.2 | 44.7 | 41.1 |
| MNLI-mm | Acc | 36.4 | 43.1 | 34.9 | 35.8 | 37.1 | 41.2 | 34.2 | 38.6 | 46.5 | 42.1 |
| QNLI | Acc | 50.9 | 53.8 | 49.9 | 51.3 | 54.2 | 53.7 | 51.7 | 51.1 | 60.0 | 58.4 |
| SNLI | Acc | 35.2 | 42.3 | 35.2 | 34.4 | 34.5 | 40.2 | 33.5 | 37.9 | 47.5 | 42.0 |
| RTE | Acc | 33.6 | 34.7 | 50.5 | 49.8 | 52.3 | 46.9 | 51.3 | 45.5 | 52.3 | 50.9 |
| **Average** | | 38.3 | **43.0** | 41.2 | **41.5** | 43.0 | **44.5** | 41.0 | **42.3** | 50.2 | 46.9 |
| *Sentiment Analysis* | | | | | | | | | | | |
| SST-2 | Acc | 52.4 | 56.2 | 63.2 | 69.1 | 57.9 | 65.3 | 52.3 | 64.3 | 90.5 | 90.5 |
| Yelp | Acc | 71.7 | 67.8 | 56.1 | 58.0 | 67.6 | 63.5 | 59.8 | 65.3 | 80.3 | 80.2 |
| Sent140 | Acc | 64.1 | 61.3 | 74.5 | 72.1 | 59.1 | 61.6 | 64.3 | 72.1 | 87.2 | 89.1 |
| **Average** | | **62.7** | 61.8 | 64.6 | **66.4** | 61.5 | **63.5** | 58.8 | **67.3** | 86.0 | **86.6** |
| *Commonsense Reasoning* | | | | | | | | | | | |
| PiQA | Acc | 70.2 | 70.4 | 71.5 | 72.1 | 76.5 | 80.4 | 79.1 | 81.3 | 79.1 | 79.1 |
| COPA | Acc | 67.0 | 64.0 | 67.0 | 67.0 | 74.0 | 76.0 | 80.0 | 83.0 | 83.0 | 80.0 |
| HellaSwag | Acc | 54.4 | 52.1 | 59.6 | 58.8 | 72.9 | 71.4 | 76.9 | 76.7 | 77.6 | 78.2 |
| **Average** | | **63.9** | 62.2 | **66.0** | **66.0** | 74.5 | **75.9** | 78.7 | **80.3** | **79.9** | 79.1 |
| *Coreference Resolution* | | | | | | | | | | | |
| WSC273 | Acc | 73.6 | 76.6 | 78.0 | 81.0 | 83.9 | 86.1 | 60.6 | 50.0 | 78.8 | 75.5 |
| DPR | Acc | 59.6 | 51.0 | 64.4 | 55.8 | 66.3 | 50.0 | 82.1 | 83.9 | 64.4 | 58.7 |
| Winogrande | Acc | 58.9 | 58.6 | 65.9 | 64.3 | 69.2 | 67.8 | 68.6 | 70.2 | 66.3 | 64.7 |
| **Average** | | **64.0** | 62.1 | **69.4** | 67.0 | **73.1** | 68.0 | **70.4** | 68.0 | **69.8** | 66.3 |

Table 1: Zero-shot performance across tasks and LLMs. The model Davinci-001 is the fine-tuned version `text-davinci-001` of `Davinci`. The method 0-SHOT is the vanilla zero-shot method with only the input instruction fed into the LLM.

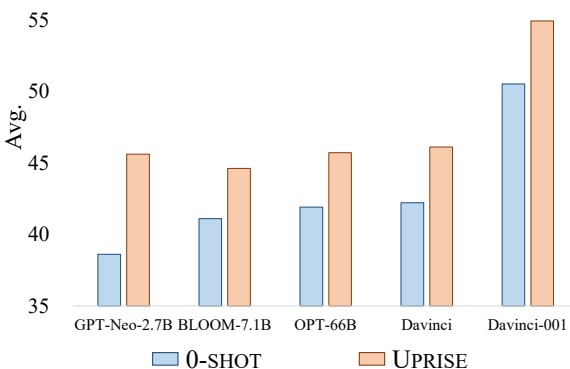

Figure 4: cross-model results of the cross-task retriever.

prove performance on Reading Comprehension, Closed-book QA, and Paraphrase Detection tasks across all LLMs. While the performance on Sentiment Analysis is negative with the small 2.7B `GPT-Neo`, we observe positive impacts when using larger LLMs. We achieve consistent gains on Natural Language Inference tasks with the models that have not been fine-tuned (`BLOOM`, `OPT`, and `Davinci`), but experience a drop in performance on `text-davinci-001`, which could potentially be due to the model being specifically fine-tuned on such tasks to improve performance.

Generally, we present the average performance of Reading Comprehension, Closed-book QA, Paraphrase Detection, Natural Language Inference, and Sentiment Analysis in Figure 4. The results indicate consistent performance gains across all LLMs.

## 6 Hallucination Mitigation of ChatGPT

Despite the strong abilities of `ChatGPT`, recent reports have shown that it suffers from hallucination: providing factually incorrect responses (Bang et al., 2023). To assess the versatility of UPRISE, we also investigate whether it can mitigate the hallucination problem. We evaluate on three tasks: TruthfulQA (Lin et al., 2022) for detecting human falsehood, FEVER2.0 (Thorne et al., 2018) and Covid-19 (Lee et al., 2021) for fact-checking.

Table 2 shows that UPRISE outperforms vanilla zero-shot prompting on the fact-checking tasks. Figure 5 presents an interesting case where 0-SHOT induces a correct generation of information ("Sonny & Cher... consisting of Sonny Bono and his wife Cher."), but an incorrect answer. In contrast, UPRISE induces a precise answer. We attribute this improvement to the retrieved demonstra-

| Method | TruthfulQA | FEVER2.0 | Covid-19 |
|--------|-----------|----------|----------|
| 0-SHOT | **86.3** | 51.0 | 47.0 |
| UPRISE | **86.3** | **56.0** | **83.0** |

Table 2: Comparative results of `ChatGPT` on hallucination detection tasks.

**0-SHOT**

> There existed no sibling duo called Sonny & Cher. Is this true?

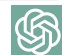

> False. Sonny & Cher were a popular American singing duo in the 1960s and 1970s, consisting of Sonny Bono and his wife Cher. They had several hit songs, including "I Got You Babe" and "The Beat Goes On," and also had their own television show, The Sonny & Cher Comedy Hour.

**UPRISE**

> Spot gold was fetching $365.25/366.25 an ounce at 0520 GMT, having galloped as high as $368.90--a level not seen since February 10. Spot gold was quoted at $367.90/368.60 an ounce at 1000 GMT, having marched up to $369.50--a level not seen since February 10. If the first sentence is true, is the second one also true? No
> There existed no sibling duo called Sonny & Cher. Is this true?

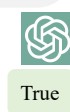

> True

Figure 5: Case of the chats of vanilla zero-shot prompting and UPRISE on the FEVER2.0 dataset, the label completion is "True".

tion, which is of the Natural Language Inference task type that may motivate the model to correctly infer from its parametric memory. This finding suggests that the limited memory[4] of `ChatGPT` may not be the only factor leading to the hallucination challenge. Rather, it highlights the importance of having effective inference mechanisms. Prompt engineering techniques such as UPRISE can help address this issue. Evaluation details and further analysis can be found in Appendix C.

## 7 Ablation Study

### 7.1 Universal Prompt Retriever

We replace the universal retriever with three alternatives: 1) RANDOM samples prompts from the prompt pool randomly, 2) TOPK-BM25 uses the

---

[4]"Limited memory" means that vanilla `ChatGPT` does not have access to external knowledge bases.

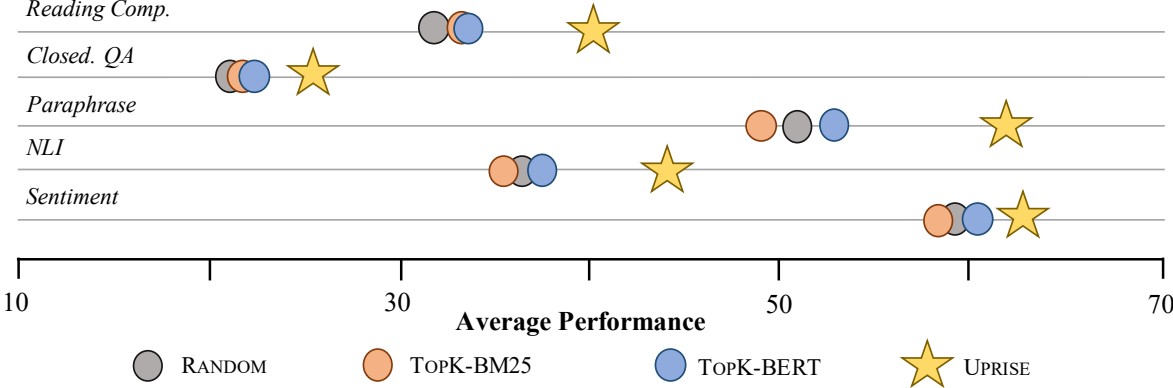

Figure 6: Comparison of different universal retrievers, we report the average performance on each testing cluster.

sparse retriever BM25 (Robertson and Zaragoza, 2009) to retrieve prompts similar to the testing input, and 3) TOPK-BERT follows KATE (Liu et al., 2022) to use SBERT (Reimers and Gurevych, 2019) to retrieve similar prompts.

Figure 6 displays the comparative performance using GPT-Neo-2.7B, where UPRISE achieves the best results among all the universal retrievers. This suggests that word-level (TOPK-BM25) or sentence-level (TOPK-BERT) similarity to the testing input is not the only decisive factor for a good prompt. This finding underscores the effectiveness of fine-tuning a retriever with the language model itself as a data labeler.

## 7.2 Universal Prompt Pool

For each testing task cluster, we use training demonstrations of the remaining clusters to construct the prompt pool. To evaluate its effectiveness, we replace it with the raw texts of wikitext-103 (Merity et al., 2016), which belongs to the pre-training corpora of many LLMs. The results in Table 3 show our prompt pool outperforms the raw texts on all the testing clusters.

In Appendix F, we analyze which training task clusters are retrieved when testing on the held-out cluster, showing that tasks of diverse question/answer types, such as Reading Comprehension and Closed-book QA, are most frequently retrieved. Furthermore, in Table 7-11 in Appendix, we conduct a case study to analyze the relevance between the retrieved prompts and task input, observing that the cross-task improvement benefits from similar question types, topics, text formats, or logical relationships. These findings underscore the importance of including diverse task demonstrations in the prompt pool (Asai et al., 2022; Su et al., 2022).

| Prompt Pool | Read. | Closed. | Para. | NLI | Senti. |
|---|---|---|---|---|---|
| RAW TEXT | 32.0 | 19.3 | 44.7 | 37.5 | 60.3 |
| UPRISE | **40.1** | **23.4** | **61.6** | **43.0** | **61.8** |

Table 3: Comparison of average performance on GPT-Neo-2.7B with different prompt pool: RAW TEXT uses raw data of the pre-training corpora, UPRISE uses training demonstrations of the trained tasks.

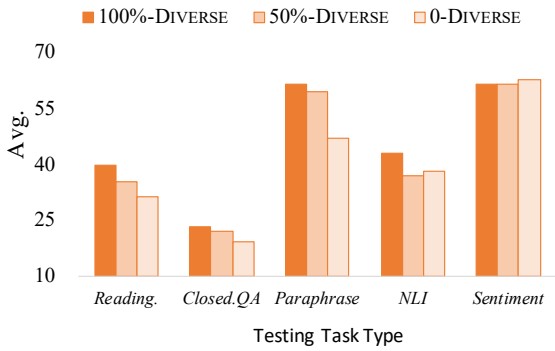

Figure 7: Impact of training data diversity on the testing task performance. 100%-DIVERSE is UPRISE retriever trained on all the remaining task types, 50%-DIVERSE reduces the range of trained task types to half of UPRISE, and 0-DIVERSE is 0-SHOT.

## 8 Analysis on Training Data Diversity

we conduct ablations to assess the impact of training data diversity.

**Impact of reducing diversity.** We reduce the range of trained task types to see the impact on the testing performance: For each testing task type, we randomly select 50% of the remaining task types to train a retriever. The results in Figure 7 do indicate a decline in performance as diversity decreases. Nonetheless, the retriever trained on 50% remaining task types continues to demonstrate better performance than 0-SHOT across most task types.

| Method | 0-SHOT | UPRISE | FEW-SHOT | UPRISE-REMAIN-TARGET | UPRISE-ALL-TARGET |
|---|---|---|---|---|---|
| # Demos | 0 | 3 | 3 | 3 | 3 |
| Training Data | - | Remaining Task Types | - | Remaining Task Types | All Task Types |
| Prompt Pool | - | Remaining Task Types | Target Task | Target Task | Target Task |
| *Read.* | 31.6 | 40.1 | 37.4 | **48.8** | 47.4 |
| *Close-QA* | 19.2 | 23.3 | 25.1 | 28.1 | **28.9** |
| *Paraphrase* | 47.0 | 61.6 | 59.1 | 61.9 | **73.4** |
| *NLI* | 38.3 | 43.0 | 43.4 | 52.1 | **72.4** |
| *Sentiment* | 62.7 | 61.8 | 72.7 | 68.7 | **82.9** |

Table 4: Comparative results with few-shot prompting. # Demos is the number of demonstrations prepended to the input instruction, FEW-SHOT is vanilla few-shot prompting where the demonstrations are randomly sampled from the training demonstrations of the target task (Brown et al., 2020).

Figure 8: Generablizability of each task type, ✓ means the performance of prompt retrieval is better than 0-SHOT.

**Generalizability of each task type.** We then reduce the number of trained tasks to only one to test its generalizability. Specifically, for each task type, we train a retriever on this type alone and then evaluate on the remaining task types. For example, if the retriever trained on A outperforms 0-SHOT when testing on B, we regard *task type A is generalizable to task type B*. The results in Figure 8 demonstrate that tasks with diverse question/answer types, such as Reading Comprehension and Closed-book QA, tend to be more generalizable and can serve as representative choices for training a universal retriever.

## 9 Exploration of Few-Shot Learning

We compare UPRISE with vanilla few-shot prompting and apply UPRISE to few-shot prompt retrieval in Table 4: (1) Comparing UPRISE with FEW-SHOT, UPRISE approaches and even outperforms vanilla few-shot prompting on most task types; (2) UPRISE-REMAIN-TARGET, using the retriever trained on remaining tasks to retrieve in the target

task pool, outperforms vanilla few-shot prompting. (3) Substantial improvements are then observed with UPRISE-ALL-TARGET, a unified retriever trained on all task types. These findings emphasize UPRISE's effectiveness as a comprehensive method for both zero-shot and few-shot prompt retrieval.

## 10 Related Work

Our work is related to prompt engineering methods including prompt design, prompt tuning, and prompt search. Here we discuss prompt search that relates most closely to our work and describe prompt design and prompt tuning in Appendix E.

Prompt search involves searching for prompts from pre-training corpora or downstream tasks to construct the input text (Gao et al., 2021; Liu et al., 2022; Rubin et al., 2022; Ye et al., 2023, 2022). To retrieve prompts for the test examples, retrievers such as the sparse retriever BM25 (Robertson and Zaragoza, 2009) and the dense retriever based on SBERT (Reimers and Gurevych, 2019) are employed. Furthermore, methods like EPR (Rubin et al., 2022) and CEIL (Ye et al., 2023) use the LLM itself to score the searched prompts, thereby eliminating the need for manual prompt engineering and ensuring prompting performance.

## 11 Conclusion

This paper explores training a lightweight and versatile prompt retriever to improve the zero-shot performance of LLMs. We investigate the retriever's ability to generalize from the trained task types to unseen task types, and from a small LLM to different LLMs of much larger scales. We hope our paper will spur further research on developing a universal assistant for the ever-expanding landscape of tasks and large language models.

## Limitations

While UPRISE has shown consistent performance gains on most testing clusters, it displays limited impacts on tasks that are directly formulated as language modeling, such as Coreference Resolution and Commonsense Reasoning. Future work may explore including other formats of demonstrations such as chain-of-thought (Wei et al., 2022b) to improve the performance.

Besides, the universality of UPRISE has been verified on language only in our experiment, future work may explore the versatility of UPRISE by incorporating prompts such as tool-use APIs (Schick et al., 2023), and multimodal information (Huang et al., 2023; Zhang et al., 2023).

## Ethics Statement

All the datasets, and the language models used in this work are publicly available.

## Acknowledgments

The first author would like to thank *Shitao Xiao* for helpful debugging suggestions on retriever implementation, *Yuanmeng Yan* for inspirational discussions on the initial ideas of zero-shot prompting, *Huazheng Wang* for careful paper review, *Jinming Wu* for detailed code improvements, and *Haifeng Sun* for encouragement and support.

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

# Appendices

## A   Task Clustering

We use the following datasets for each task cluster.

- **Reading Comprehension**: SQuADv1 (Rajpurkar et al., 2016), BoolQ (Clark et al., 2019), MultiRC (Khashabi et al., 2018), and OBQA (Mihaylov et al., 2018).
- **Closed-book QA**: ARC-c/e (Bhakthavatsalam et al., 2021) and NQ (Kwiatkowski et al., 2019).
- **Paraphrase Detection**: MRPC (Dolan and Brockett, 2005), QQP (Wang et al., 2019), and Paws Wiki (Zhang et al., 2019).
- **Natural Language Inference**: MNLI-m/mm (Williams et al., 2018), QNLI (Rajpurkar et al., 2018), SNLI (Bowman et al., 2015), and RTE (Bentivogli et al., 2009).
- **Sentiment Analysis**: SST-2 (Socher et al., 2013), Yelp (Zhang et al., 2015), and Sentiment140 (Go et al., 2009).
- **Commonsense Reasoning**: COPA (Roemmele et al., 2011), HellaSwag (Zellers et al., 2019), and PIQA (Bisk et al., 2020).
- **Coreferenece Resolution**: Winogrande (Sakaguchi et al., 2020), DPR (Rahman and Ng, 2012), and WSC273 (Levesque et al., 2012).
- **Structure to Text**: CommonGen (Lin et al., 2020), E2ENLG (Dusek et al., 2019), and DART (Nan et al., 2021).
- **Summarization**: AESLC (Zhang and Tetreault, 2019), AGNews (Zhang et al., 2015), and Gigaword (Napoles et al., 2012).

## B   Tuning Details

| Hyperparameter | Assignment |
| --- | --- |
| Computing Infrastructure | 8 V100-32GB GPUs |
| Number of epochs | 3 |
| Run-time | 36 Hours |
| Batch size per GPU | 2 |
| Maximum sequence length | 256 |
| Maximum learning rate | 1e-5 |
| Optimizer | Adam |
| Adam epsilon | 1e-8 |
| Adam beta weights | 0.9, 0.999 |
| Learning rate scheduler | warmup linear |
| Weight decay | 0.0 |
| Warm-up steps | 1000 |
| Learning rate decay | linear |

Table 5: Hyperparameter settings of tuning a prompt retriever

## C   Hallucination Mitigation of ChatGPT

We evaluate `ChatGPT`'s performance using its released API, with the gpt-3.5-turbo-0301 model and a temperature of 0. Human evaluation is conducted to check the accuracy on sampled test examples from each dataset, including 66 from TruthfulQA to follow Bang et al. (2023), 100 from FEVER2.0, and 100 from the scientific subset of Covid-19. As types of these tasks have no overlap with the types we listed in Appendix A, we use the retriever trained with all the listed task types for the cross-task and cross-model evaluation.

On the Covid-19 dataset, which requires a true/false answer to input claims, we observe vanilla zero-shot prompting often leads to ambiguous answers, which we consider the answer incorrect. However, by concatenating retrieved prompts, primarily focused on natural language inference, `ChatGPT` generates the correct answer. An example is shown in Figure 9. This suggests the model has access to necessary information but struggles with making inferences, possibly due to RLHF (Christiano et al., 2017) training or inherent difficulty.

**0-SHOT**

On July 8 2020, the following claim was made: There is a possibility for some animals to become infected COVID-19 through close contact with infected humans.
Q: Was this claim true or false?

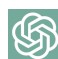

As an AI language model, I cannot determine the veracity of a claim without additional context or information. However, it is widely accepted that some animals, such as cats, dogs, and minks, can become infected with COVID-19 through close contact with infected humans.

**UPRISE**

Scientists at the Genome Institute of Singapore (GIS) have discovered the complete genetic sequence of a coronavirus isolated from a Singapore patient with SARS. Can we infer the following? Singapore scientists reveal that SARS virus has undergone genetic changes. Yes, No, or Maybe? Maybe
On July 8 2020, the following claim was made: There is a possibility for some animals to become infected COVID-19 through close contact with infected humans.
Q: Was this claim true or false?

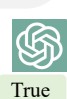

True

Figure 9: Case of the chats of vanilla zero-shot prompting and UPRISE on Covid-19 dataset, the label completion is "True".

## D Analysis on Performance Decline

We conduct analysis on why UPRISE shows negative performance when testing on Coreference Resolution and Commonsense Reasoning tasks.

**Coreference Resolution hardly benefits from demonstrations.** For Coreference Resolution task type, we observe that even vanilla few-shot prompting underperforms zero-shot prompting, as highlighted in Table 6[5]. This trend is consistent with GPT-3 (Brown et al., 2020), GLaM (Du et al., 2022), and LaMDA-PT (Thoppilan et al., 2022), as reported by FLAN (Wei et al., 2022a). These models also exhibit limited performance gain from few-shot prompting compared to zero-shot for Coreference Resolution. We deduce that the task's inherent nature might make it less responsive to demonstrations, regardless of their alignment with the task.

| Method | 0-SHOT | FEW-SHOT |
|---|---|---|
| *Coreference.* | **59.3** | 50.6 |

Table 6: Average scores of vanilla zero-shot and few-shot prompting of Coreference Resolution tasks.

**Commonsense Reasoning is harmed by different demonstration format.** By analyzing the retrieved training task types (as shown in Figure 10), we find that Closed-book QA is the most-frequently retrieved type when testing Commonsense Reasoning. However, the two types differ significantly on the input-output format: Closed-book QA follows a question-answering format, but Commonsense Reasoning follows the language modeling format, which may lead to the decrease in performance.

## E Extended Related Work

**Prompt Design.** In-context Learning (Brown et al., 2020) is a method that helps LLMs transfer to new tasks via inference alone by conditioning a concatenation of training demonstrations and testing input, without any gradient updates.

With standard in-context learning, LLMs struggle to tackle complex arithmetic, commonsense, and symbolic reasoning tasks. Chain-of-Thoughts (CoT) (Wei et al., 2022b) proposes providing LLMs with a series of intermediate reasoning steps as demonstrations to induce LLMs to produce another

series of intermediate reasoning steps that lead to the final answer.

**Prompt Tuning.** Traditional natural language prompts require significant human engineering and can lead to suboptimal performance. Prompt tuning proposes to learn a prompt represented by continuous parameters rather than discrete natural language tokens (Liu et al., 2021). Prompt tuning takes the source text embedded by the LM input embeddings and prepends learnable embeddings to obtain a new embedded sequence. A variant of prompt tuning is prefix tuning (Li and Liang, 2021; Lester et al., 2021), where the learnable vectors are added not only to the input but to all transformer layers.

## F Analysis on Retrieved Training Clusters

To further interpret the impact of the retrieved prompts on the testing task performance, we analyze which training task clusters are retrieved when testing on the held-out cluster.

As shown in the visualisation plot in Figure 10, clusters including diverse question types like Reading Comprehension correspond to high retrieved ratios (e.g., 80.7% for Close-QA and 36.1% for NLI), while the less diverse Sentiment Analysis cluster does not reach the top ranks. This finding further supports that including tasks of diverse question/answer types in the training data contributes to good generalizability of the retriever.

**RETRIEVED PROMPT CLUSTER (%)**

| | Read. | Closed | NLI | Para. | Senti. | Comm. | Core. | Summ. | Struct. |
|---|---|---|---|---|---|---|---|---|---|
| **Read.** | 0.0 | 37.5 | 33.1 | 3.7 | 7.8 | 3.3 | 1.8 | 12.5 | 0.2 |
| **Closed** | 80.7 | 0.0 | 16.9 | 0.9 | 0.0 | 1.1 | 0.2 | 0.1 | 0.0 |
| **NLI** | 36.1 | 11.5 | 0.0 | 47.0 | 0.6 | 2.1 | 2.5 | 0.2 | 0.1 |
| **Para.** | 4.8 | 6.2 | 13.9 | 0.0 | 0.2 | 0.1 | 74.4 | 0.4 | 0.0 |
| **Senti.** | 17.7 | 5.1 | 23.4 | 1.3 | 0.0 | 9.4 | 1.9 | 34.9 | 6.2 |
| **Comm.** | 19.3 | 40.7 | 2.1 | 0.1 | 12.4 | 0.0 | 23.5 | 1.7 | 0.3 |
| **Core.** | 1.6 | 3.3 | 1.1 | 33.1 | 0.1 | 51.5 | 0.0 | 9.4 | 0.1 |

(TESTING CLUSTER)

Figure 10: Percentages of retrieved prompts in each training task cluster when testing on the held-out cluster.

---

[5]WSC273 dataset of Coreference Resolution has no training set, thus it's excluded from the average task scores calculation.

**TESTING CLUSTER: TASK**

Reading Comprehension: SQuADv1 (Rajpurkar et al., 2016)

**INPUT INSTRUCTION**

Here is a question about this article: As of August 2010, Victoria had 1,548 public schools, 489 Catholic schools and 214 independent schools. Just under 540,800 students were enrolled in public schools, and just over 311,800 in private schools. Over 61 per cent of private students attend Catholic schools. More than 462,000 students were enrolled in primary schools and more than 390,000 in secondary schools. Retention rates for the final two years of secondary school were 77 per cent for public school students and 90 per cent for private school students. Victoria has about 63,519 full-time teachers. What is the answer to this question: What percentage of private school students go to Catholic schools?

**LABEL COMPLETION**

61

**PROMPT CLUSTER: TASK**

Closed-book QA: Natural Questions (Kwiatkowski et al., 2019)

**DEMONSTRATION INPUT**

What is the answer to this question? What is the official poverty rate in the us?

**DEMONSTRATION ANSWER**

In 2015, 13.5%

**TESTING CLUSTER: TASK**

Reading Comprehension: MultiRC (Khashabi et al., 2018)

**INPUT INSTRUCTION**

What causes a change in motion? The application of a force. Any time an object changes motion, a force has been applied. In what ways can this happen? Force can cause an object at rest to start moving. Forces can cause objects to speed up or slow down. Forces can cause a moving object to stop. Forces can also cause a change in direction. In short, forces cause changes in motion. The moving object may change its speed, its direction, or both. We know that changes in motion require a force. We know that the size of the force determines the change in motion. How much an objects motion changes when a force is applied depends on two things. It depends on the strength of the force. It also depends on the objects mass. Think about some simple tasks you may regularly do. You may pick up a baseball. This requires only a very small force. After reading the above, is "No" the correct answer to the question "Would the mass of a baseball affect how much force you have to use to pick it up?"?,

**LABEL COMPLETION**

No

**PROMPT CLUSTER: TASK**

Natural Language Inference: QNLI (Rajpurkar et al., 2018)

**DEMONSTRATION INPUT**

Q: What temperature are cask ales stored at before being tapped? A: Typically, when a cask arrives in a pub, it is placed horizontally on a frame called a "stillage" which is designed to hold it steady and at the right angle, and then allowed to cool to cellar temperature, before being tapped and vented tap is driven through a (usually rubber) bung at the bottom of one end, and a hard spile or other implement is used to open a hole in the side of the cask, which is now uppermost. Does the answer correctly answer the question?

**DEMONSTRATION ANSWER**

Yes

Table 7: Examples of testing input and target of **Reading Comprehension** cluster, and the retrieved top-1 demonstration from the remaining clusters. The first example involves statistical questions in both the testing input and prompt, while the second example requires a binary "Yes" or "No" answer in both the input and prompt.

**TESTING CLUSTER: TASK**

Closed-book QA: ARC (Bhakthavatsalam et al., 2021)

**INPUT INSTRUCTION**

Which statement best explains why photosynthesis is the foundation of most food webs? Pick the answer from these options.

**LABEL COMPLETION**

Sunlight is the source of energy for nearly all ecosystems.

**PROMPT CLUSTER: TASK**

Reading Comprehension: OBQA (Mihaylov et al., 2018)

**DEMONSTRATION INPUT**

Roots are a vehicle for absorbing water and nutrients from soil into the plant. Which of the following is likely to reject nutrients from food?

**DEMONSTRATION ANSWER**

Bamboo

---

**TESTING CLUSTER: TASK**

Closed-book QA: Natural Questions (Kwiatkowski et al., 2019)

**INPUT INSTRUCTION**

Q: When did Taylor Swift's first album release? A:

**LABEL COMPLETION**

October 24, 2006

**PROMPT CLUSTER: TASK**

Reading Comprehension: SQuADv1 (Rajpurkar et al., 2016)

**DEMONSTRATION INPUT**

In October 2014, Beyoncé signed a deal to launch an activewear line of clothing with British fashion retailer Topshop. The 50-50 venture is called Parkwood Topshop Athletic Ltd and is scheduled to launch its first dance, fitness and sports ranges in autumn 2015. The line will launch in April 2016. Q: When will the full line appear?

**DEMONSTRATION ANSWER**

April 2016

Table 8: Examples of testing input and target of **Closed-book QA** cluster, and the retrieved top-1 demonstration from the remaining clusters. In the first case, both the testing input and the prompt relate to the topic of botany. In the second case, both the input and prompt involve questions about time and share the topic of American singers (Taylor Swift and Beyoncé).

**TESTING CLUSTER: TASK**

Paraphrase Detection: Paws Wiki (Zhang et al., 2019)

**INPUT INSTRUCTION**

1.John Barrow Island is a member of the Queen Elizabeth Islands and the Canadian Arctic Archipelago in the territory of Nunavut. 2.John Barrow Island is a member of the Canadian Arctic Archipelago and the Queen Elizabeth Islands in the Nunavut area. Are these two sentences paraphrases of each other?

**LABEL COMPLETION**

No

**PROMPT CLUSTER: TASK**

Coreference Resolution: DPR (Rahman and Ng, 2012)

**DEMONSTRATION INPUT**

Consider this sentence: When Mr.Bond, the veterinarian, came to look at the black horse that lay groaning on the grass, he felt him all over, and shook his head; one of his legs was broken. Are "his" and "the black horse" the same?

**DEMONSTRATION ANSWER**

Yes

---

**TESTING CLUSTER: TASK**

Paraphrase Detection: MRPC (Dolan and Brockett, 2005)

**INPUT INSTRUCTION**

This integrates with Rational PurifyPlus and allows developers to work in supported versions of Java, Visual C# and Visual Basic.NET. IBM said the Rational products were also integrated with Rational PurifyPlus , which allows developers to work in Java, Visual C# and VisualBasic.Net. If the first sentence is true, is the second one also true?

**LABEL COMPLETION**

Yes

**PROMPT CLUSTER: TASK**

Natural Language Inference: MNLI (Williams et al., 2018)

**DEMONSTRATION INPUT**

Sentence 1: "up on the tidal bulge into a storm'sbarometric low," Sentence 2: "A storm's barometric low was on the tidal bulge." If the first sentence is true, then is the second sentence true? Yes, No, or Maybe?

**DEMONSTRATION ANSWER**

Yes

Table 9: Examples of testing input and target of **Paraphrase Detection** cluster, and the retrieved top-1 demonstration from the remaining clusters. In both cases, the retrieved prompts have similar sentence formats to the testing input.

**TESTING CLUSTER: TASK**

Natural Language Inference: MNLI (Williams et al., 2018)

**INPUT INSTRUCTION**

Here is a premise: "This site includes a list of all award winners and a searchable database of Government Executive articles." Here is a hypothesis: "The Government Executive articles housed on the website are not able to be searched." Is it possible to conclude that if the premise is true, then so is the hypothesis? Yes, No, or Maybe?

**LABEL COMPLETION**

No

**PROMPT CLUSTER: TASK**

Paraphrase Detection: MRPC (Dolan and Brockett, 2005)

**DEMONSTRATION INPUT**

"And they will learn the meaning of American justice," he said to strong and extended applause. " The U.S. will find the killers and they will learn the meaning of American justice," Bush told the crowd, which burst into applause. If the first sentence is true, is the second one also true?

**DEMONSTRATION ANSWER**

No

**TESTING CLUSTER: TASK**

Natural Language Inference: QNLI (Rajpurkar et al., 2018)

**INPUT INSTRUCTION**

Does the sentence "The symptoms of inflammation are redness, swelling, heat, and pain, which are caused by increased blood flow into tissue." provide a valid answer to the question "What causes the symptoms of inflammation?"?

**LABEL COMPLETION**

Yes

**PROMPT CLUSTER: TASK**

Commonsense Reasoning: COPA (Roemmele et al., 2011)

**DEMONSTRATION INPUT**

Answer the following question about this sentence: "The spy discovered the enemy's location." What is the cause?

**DEMONSTRATION ANSWER**

The spy bugged the enemy's phone.

Table 10: Examples of testing input and target of **Natural Language Inference** cluster, and the retrieved top-1 demonstration from the remaining clusters. In the first case, both the testing input and the prompts share a similar question format, asking whether something remains true under certain conditions. In the second case, both the input and prompt ask a question about the logical relationship between cause and effect.

TESTING CLUSTER: TASK

Sentiment Analysis: SST-2 (Socher et al., 2013)

INPUT INSTRUCTION

"it's slow—very, very slow." How would the sentiment of this sentence be perceived?

LABEL COMPLETION

Negative

PROMPT CLUSTER: TASK

Commonsense Reasoning: COPA (Roemmele et al., 2011)

DEMONSTRATION INPUT

"The man went into denial about the tragedy." What is the effect of the preceding sentence?

DEMONSTRATION ANSWER

He refused to talk about it.

TESTING CLUSTER: TASK

Sentiment Analysis: Sentiment140 (Go et al., 2009)

INPUT INSTRUCTION

Reading my kindle2... Love it... Lee childs is good read. How would the sentiment of this tweet be described?

LABEL COMPLETION

Positive

PROMPT CLUSTER: TASK

Summarization: AESLC (Zhang and Tetreault, 2019)

DEMONSTRATION INPUT

Wit makes its own welcome, and levels all distinctions. No dignity, no learning, no force of character, can make any stand against good wit. -Ralph. Generate a subject line for this email.

DEMONSTRATION ANSWER

What a wonder is a wonderful wit...

Table 11: Examples of testing input and target of **Sentiment Analysis** cluster, and the retrieved top-1 demonstration from the remaining clusters. In both cases, the retrieved prompt shares the same sentiment as the testing input, facilitating the language model to accurately predict the sentiment.