# OpenReview forum: "UPRISE: Universal Prompt Retrieval for Improving Zero-Shot Evaluation"
_EMNLP/2023/Conference — EMNLP 2023 Main_

### Official Review · Reviewer_xPye · 2023-08-05

**Soundness:** 4

**Excitement:**

3: Ambivalent: It has merits (e.g., it reports state-of-the-art results, the idea is nice), but there are key weaknesses (e.g., it describes incremental work), and it can significantly benefit from another round of revision. However, I won't object to accepting it if my co-reviewers champion it.

**Paper Topic And Main Contributions:**

The introduction of UPRISE (Universal Prompt Retrieval for Improving Zero-Shot Evaluation), a method that utilizes a lightweight and versatile retriever to automatically retrieve prompts. This approach aids in the zero-shot evaluation of LLMs.

The design of a two-step process in UPRISE: prompt retrieval and response generation.

A comprehensive evaluation of UPRISE, where it demonstrates superior performance in comparison to traditional prompt engineering methods.

An analysis of how and why UPRISE works, including a detailed examination of the retrieval process and its impact on zero-shot performance.

**Questions For The Authors:**

Question A: Could you elaborate on the training data used for UPRISE? How diverse and comprehensive was it, and how might the quality of the training data impact the performance of UPRISE?

Question B: How well might the UPRISE method generalize to other learning settings, beyond zero-shot evaluation? Could the method be adapted for few-shot learning or fine-tuning settings?

Question C: Were any real-world scenarios or complex tasks used to test the UPRISE method? If not, how do you anticipate the method would perform in such contexts?

Question D: What are the computational requirements for the UPRISE method? Given the mention of a lightweight and versatile retriever, can you provide more details about its efficiency in comparison to traditional methods?

Question E: Could the UPRISE method be applied to language models other than LLMs? If so, how might the performance compare?

Question F: How sensitive is UPRISE to the choice of retrieval algorithm? Could different algorithms significantly impact the performance?

**Reasons To Accept:**

Novel Approach: The paper introduces UPRISE, a novel method that addresses a significant challenge in the field of language models: the need for task-specific prompt engineering. This is a unique approach that could potentially improve the efficiency of LLMs.

Comprehensive Evaluation: The paper provides a comprehensive evaluation of the proposed method, demonstrating its effectiveness in comparison to traditional prompt engineering methods. This provides strong evidence for the effectiveness of UPRISE.

Detailed Analysis: The authors provide a thorough analysis of how and why UPRISE works, which not only validates their approach but also contributes to a deeper understanding of the underlying mechanisms. This could be valuable for future research in this area.

**Reasons To Reject:**

Dependent on Quality of Training Data: The effectiveness of UPRISE might be largely dependent on the quality and diversity of prompts in the training data. If the training data is not diverse or rich enough, the model might not generalize well to unseen prompts.

Limited to Zero-Shot Learning: The focus of this paper is specifically on zero-shot evaluation. If the conference is looking for papers that address a broader range of learning settings, this might not be the best fit.

Need for More Real-World Testing: While the paper demonstrates the effectiveness of UPRISE through experiments, it may not have sufficiently demonstrated how this method performs in real-world scenarios or with complex tasks. More testing might be needed to confirm the practical applicability of UPRISE.

**Reproducibility:**

3: Could reproduce the results with some difficulty. The settings of parameters are underspecified or subjectively determined; the training/evaluation data are not widely available.

**Reviewer Confidence:**

4: Quite sure. I tried to check the important points carefully. It's unlikely, though conceivable, that I missed something that should affect my ratings.

---

> ### Author Rebuttal · Authors · 2023-08-28
>
> Thanks for your detailed and comprehensive review, we hope the following answers can address your concerns, and the analyses and clarifications will be incorporated into our paper.
>
> ## Q1: Analysis of quality and diversity of training Data
> ### Q1.1 Could you elaborate on the training data? How diverse and comprehensive was it?
> We appreciate your concern and offer the following clarifications.
>
> Our training data encompasses various task types, such as Reading Comprehension, Closed-book QA, Paraphrase Detection, and Natural Language Inference (line 305-311). Each task type includes at least 3 different tasks, amounting to 30 task datasets in total (Appendix A). Then, each task is represented by 7 templates to create prompts (line 167-171), resulting a prompt pool of 180k examples for each testing task (line 317-320). The choice of task clustering and settings follows the well-regarded FLAN approach for multi-task learning and strong zero-shot generalizability.
>
> ### Q1.2 The effectiveness might be largely dependent on the diversity, how might it impact the performance?
> Yes, we concur with you that diversity is very important in achieving generalizability, but I hope the following analysis can alleviate you concern that "If the training data is not diverse or rich enough, the model might not generalize well to unseen prompts."
>
> To evaluate impact of training data diversity, we conduct evaluations by reducing the range of trained task types: For each testing task type, we randomly select 50% of the remaining task types to train a retriever. The results, shown below, do indicate a decline in performance as diversity decreases. Nonetheless, **the retriever trained on 50% remaining task types continues to demonstrate better performance than 0-SHOT** across several task types.
>
> |                               |             UPRISE            |          UPRISE-half-diverse         | 0-shot |
> |-------------------------------|-----------------------------|----------------------------|------|
> | **Training Data/Prompt Pool** | **100% Remaining Task Types** | **50% Remaining Task Types** |  **-** |
> | Reading                       |             **40.1**              |             35.6             |  31.6  |
> | Close-QA                |             **23.3**              |             22.1             |  19.2  |
> | Paraphrase                    |             **61.6**              |             59.5             |  47.0  |
> | NLI                           |             **43.0**              |             36.9             |  38.3  |
> | Sentiment                     |             61.8              |             61.7             |  **62.7**  |
>
> Moreover, we find inspiration from Reviewer zjV3's Q6. It suggests that **incorporating representative task types, such as Reading Comprehension and Closed-book QA alone, can yield good generalizability**. This offers a data-efficient means of training a retriever with broad applicability.
>
> ## Q2: How well might UPRISE generalize to other learning settings?
> **UPRISE can be generalized to few-shot learning!** Please see Q4 for Reviewer zjV3, which emphasizes UPRISE's effectiveness for few-shot prompt retrieval.
>
> ## Q3: Were any real-world scenarios or complex tasks used to test UPRISE?
> We are pleased to share that **UPRISE has been successfully applied for complex tasks in real-world scenarios**. While we're unable to provide explicit details due to commercial sensitivities, we can confirm that UPRISE has been employed to train a Chain-of-Thought (CoT) prompt retriever for real-world applications, with GPT-3.5's feedback and subsequently evaluated on GPT-4. The results were remarkably positive, surpassing the high-performing manually-crafted CoT prompting:
>
> |                     | real-app-1  | real-app-2  | real-app-3  | real-app-4  |
> |---------------------|-----------|-----------|-----------|-----------|
> | Manual Few-shot CoT | 78.5        | 75.2        | 78.1        | 68.8        |
> | UPRISE Few-shot CoT | **86.7** | **79.0** | **81.0** | **70.9** |
>
> ## Q4: What are the computational requirements? Why is UPRISE lightweight?
> The two encoders are initialized from BERT-base (line 341), and the computing infrastructure includes 8 V100-32GB GPUs (Appendix B), training a unified retriever on all the task types takes about 1 day.
>
> When we mention UPRISE's lightweight nature, **we're comparing it with methods that involve tuning large language models**. UPRISE tunes a retriever (using two BERT-base encoders) for cross-model and cross-task versatility.
>
> ## Q5: Could UPRISE be applied to language models other than LLMs?
> Yes! To address your concern, we extend our testing to **125M GPT-2 and observe favorable performance**:
>
> |        | Reading   |  Close-QA | Paraphrase |    NLI    | Sentiment |
> |--------|---------|---------|----------|---------|---------|
> | 0-SHOT |   24.1    |   14.9    |    37.7    |   34.7    | **60.3** |
> | UPRISE | **28.3** | **15.4** |  **62.8** | **42.9** |   59.4    |
>
> ## Q6: How sensitive is UPRISE to the choice of retrieval algorithm?
> We conduct experiments to replace the training objective of UPRISE with the one in UDR [1], a very recently proposed prompt retrieval method. UDR employs a list-wise ranking loss function in addition to the in-batch negative loss.
>
> We train two retrievers on Reading Comprehension and Closed-book QA task types respectively, and evaluate their generalizability on the remaining task types. As shown below, even with a different training objective, the retriever demonstrates improved results compared to 0-SHOT, and the performances of the two methods do not differ much, suggesting the robustness and adaptability of UPRISE to different retrieval methods.
>
> |                   | 0-SHOT |    UPRISE-read    |   UPRISE-read-UDR  |     UPRISE-qa     |    UPRISE-qa-UDR   |
> |-------------------|------|-----------------|------------------|-----------------|------------------|
> |   **Algorithm**   |    -   | in-batch negative | list-wise rank [1] | in-batch negative | list-wise rank [1] |
> | **Training Data/Prompt Pool** |    -   |        Reading       |        Reading        |      Close-QA     |      Close-QA      |
> |        Reading       |  31.6  |         -         |          -         |       36.8        |        34.7        |
> |      Close-QA     |  19.2  |       24.1        |        24.3        |         -         |          -         |
> |     Paraphrase    |  47.0  |       49.7        |        51.4        |       51.0        |        52.2        |
> |        NLI        |  38.3  |       39.8        |        37.4        |       38.7        |        38.5        |
> |     Sentiment     |  62.7  |       63.2        |        64.8        |       63.9        |        64.8        |
>
> Reference:
>
> [1] Xiaonan Li et al. 2023. Unified demonstration retriever for in-context learning. In *ACL (1)*. Association for Computational Linguistics.

---

### Official Review · Reviewer_zjV3 · 2023-08-05

**Typos Grammar Style And Presentation Improvements:** 1. The right figure in the Figure 2 i…
**Soundness:** 3

**Excitement:**

3: Ambivalent: It has merits (e.g., it reports state-of-the-art results, the idea is nice), but there are key weaknesses (e.g., it describes incremental work), and it can significantly benefit from another round of revision. However, I won't object to accepting it if my co-reviewers champion it.

**Paper Topic And Main Contributions:**

This paper proposes an approach to retrieve prompts from a collection of tasks for unseen tasks. The idea is to train a dual encoder based prompt retriever by measuring the relevance between a task input sequence and prompts collected from other tasks. The retrieved top K prompts are then used to concatenate with the task input, and then are fed into a LLM to generate the prediction.

The proposed method is evaluated on 7 different types of tasks and show consistent improving performance over zero-shot setup on 4 types of tasks, which demonstrates the cross-task generalization capability.

**Questions For The Authors:**

Question A: For the prompt retriever, do both encoders share parameters? or do they initialize from independent BERT_base?
Question B: Is a different prompt retriever trained for each test task cluster?
Question C: In the result Table 1, for SQuAD dataset, results using UPRISE improve a lot (from 4.4 to 26.4) when using gpt-neo-2.7b, but not when using bloom-7.1 and opt-66b. Could you elaborate on why this happens?

**Reasons To Accept:**

The proposed approach can leverage the existing prompts from different tasks to a target tasks, which doesn't require any labeled data from target task. The paper identifies some interesting relationship between different tasks, such as sharing similar question types, topics, text formats and logical relationships. It is evaluated on a wide range of tasks, which shows both benefits and limitations of the proposed approach.

**Reasons To Reject:**

1. The paper claims that the proposed approach improves the zero-shot evaluation, but i think it is valuable and helpful to understand the gap between using prompts from other tasks and using prompts from the target task (i.e. the few-shot setting). Specifically, what will be the performance of using 3 demonstrations from the target task.
2. The analysis on why UPRISE does't perform well on some tasks (e.g. commonsense reasoning, coreference resolution) is limited, it will be better to have some more in-depth analysis.
3. I think the prompt pool is a critical component of UPRISE, and what are the representative and generalizable tasks to be included in the prompt pool is worth to explore. The prompt pool in the paper is formed by using the tasks from FLAN, it is not sure that tasks in this set are the best selection of tasks.

**Reproducibility:**

3: Could reproduce the results with some difficulty. The settings of parameters are underspecified or subjectively determined; the training/evaluation data are not widely available.

**Reviewer Confidence:**

4: Quite sure. I tried to check the important points carefully. It's unlikely, though conceivable, that I missed something that should affect my ratings.

---

> ### Author Rebuttal · Authors · 2023-08-28
>
> Thanks for your careful and insightful review. We reply to each one of your concerns below, and the analyses and clarifications will be incorporated into our paper.
>
> ## Q1: Do both encoders share parameters?
> The two encoders do not share parameters; each initializes independently from BERT_base.
>
> ## Q2: Is a different prompt retriever trained for each test task cluster?
> Yes, in order to ensure a **strict cross-task evaluation** of our method---the testing task type must have NO overlap with the trained types---we train a different prompt retriever for each testing task type.
>
> As stated in line 141-146, we follow the cross-task evaluation in FLAN. For example, suppose there are 4 task types (A,B,C,D), when testing on A type, the retriever trained on the remaining 3 types (B,C,D) are employed for prompt retrieval.
>
> ## Q3: Strange SQuAD score improvements in BLOOM-7.1B and OPT-66B
> The strange improvements in BLOOM-7.1B and OPT-66B are caused by the **redundant predictions**.
>
> Although we have restricted generation length and applied early stopping on `\n`, BLOOM-7.1B and OPT-66B continue to generate redundant predictions, impacting evaluations for both 0-SHOT and UPRISE. In contrast, GPT-2.7B-Neo, with its shorter predictions, avoids generating such redundancy. To quantify, GPT-2.7B-Neo's average generation length is `17.6`, while BLOOM-7.1B reaches `77.9` and OPT-66B reaches `63.4`.
>
> Following implementation of FLAN, we employ Exact Match and F1 to assess performance on SQuAD. Achieving high scores in these metrics requires not only content similarity but also **length** similarity between predictions and labels. In such condition, the redundant segments generated from BLOOM-7.1B and OPT-66B lead to unexpectedly low scores, even if the prediction contains the accurate answer.
>
> To verify the impact, we post-process the model predictions, removing redundant segments, and then re-evaluate the metric scores. The table below displays notably higher improvement of UPRISE after the removal, affirming that the redundant segments are the cause of the strange improvements.
>
> |Metric | Remove Redundant |  BLOOM-7.1B |        | OPT-66B |        |
> |--------------|--------|----------|------|-------|------|
> |              |        | 0-SHOT     | UPRISE | 0-SHOT | UPRISE |
> |  F1     |  Before     | 4.5        | 5.5    | 6.1     | 7.5 |
> |        | After      | **4.6**    | **36.4**| **6.8**    | **37.3**   |
> | Exact Match       |  Before    | 0.0        | 0.0    | 0.0     | 0.6    |
> |         |   After   | 0.0        | **18.8**  |**0.4**    | **21.3**|
>
>
> ## Q4: Gap between UPRISE and few-shot prompting
> Thanks for the valuable and helpful suggestion. We compare UPRISE with vanilla few-shot prompting (Few-SHOT), where 3 demonstrations are randomly sampled from the target task. We also explore few-shot prompting with UPRISE retrievers (UPRISE-remain-target and UPRISE-all-target):
>
>
> | Method        | 0-SHOT |   UPRISE |   Few-SHOT  | UPRISE-remain-target | UPRISE-all-target |
> |-------------------|------|-----------|-----------|--------------------|-----------------|
> | **# Demos**       |    0   |           3          |      3      |           3          |         3         |
> | **Training Data** |    -   | Remaining Task Types |      -      | Remaining Task Types |   All Task Types  |
> | **Prompt Pool**   |    -   | Remaining Task Types | Target Task |      Target Task     |    Target Task    |
> | Reading           |  31.6  |         40.1         |    37.4     |         **48.8**         |       47.4        |
> | Close-QA          |  19.2  |         23.3         |    25.1     |         28.1         |       **28.9**        |
> | Paraphrase        |  47.0  |         61.6         |    59.1     |         61.9         |       **73.4**        |
> | NLI               |  38.3  |         43.0         |    43.4     |         52.1         |       **72.4**        |
> | Sentiment         |  62.7  |         61.8         |    72.7     |         68.7         |       **82.9**        |
>
>
> Comparing UPRISE with Few-SHOT, UPRISE approaches and even outperforms vanilla few-shot prompting on most task types.
>
> Additionally, UPRISE-remain-target, using the retriever trained on remaining tasks to retrieve in the target task pool, outperforms vanilla few-shot prompting.
>
> Substantial improvements are then observed with UPRISE-all-target, a unified retriever trained on all task types.
>
> These findings emphasize UPRISE's effectiveness as a comprehensive method for both zero-shot and few-shot prompt retrieval!
>
> ## Q5: Why not good on Coreference Resolution and Commonsense Reasoning?
> We appreciate your concern and offer the following clarifications.
>
> **Coreference Resolution hardly benefits from demonstrations.** For Coreference Resolution task type, we observe that even vanilla few-shot prompting underperforms zero-shot prompting, as highlighted below. This trend is consistent with GPT-3, GLaM, and LaMDA-PT as reported in the FLAN paper. These models also exhibit limited performance gain from few-shot prompting compared to zero-shot for Coreference Resolution. We deduce that the task's inherent nature might make it less responsive to demonstrations, regardless of their alignment with the task.
>
> |                   |   0-SHOT   |     Few-SHOT  |
> |-------------------|:-----------:|:------------:|
> | Coreference       |  **59.3**  |     50.6      |
>
> (Note that the WSC273 dataset of Coreference Resolution has no training set, thus it's excluded from the average task scores calculation.)
>
> **Commonsense Reasoning is harmed by different demonstration format.** By analyzing the retrieved training task types, we find that Closed-book QA is the most-frequently retrieved type when testing Commonsense Reasoning. However, the two types differ significantly on the input-output format: Closed-book QA follows a question-answering format, but Commonsense Reasoning follows the language modeling format, which may lead to the decrease in performance.
>
> ## Q6: Not sure the tasks from FLAN are the best for prompt pool. What are the representative and generalizable tasks?
> We value your insights on prompt pool and representative tasks. We included FLAN's tasks in our prompt pool due to their proven success in strong zero-shot generalization via multi-task learning. Acknowledging your concern on task choice's impact, we conduct the following analysis:
>
> **Tasks with diverse questions/answers are representative.** As observed in Q1 for Reviewer 5iUs, task types of high diversity are frequently retrieved by others, suggesting the positive impact of task diversity on demonstration effectiveness.
>
> Furthermore, for each task type, we train a retriever on this type alone and then evaluate on the remaining task types, to test its generalizability. For example, if the retriever trained on A outperforms 0-SHOT when testing on B, we regard `task type A is generalizable to task type B`.
>
> The results below demonstrate that tasks with diverse question/answer types, such as Reading Comprehension and Closed-book QA, tend to be more generalizable and can serve as representative choices for training a universal retriever. We hope this insight can inspire future research in developing a broadly applicable retriever.
>
>
> |  | Reading | Close-QA | Paraphrase | NLI | Sentiment |
> |-------------------------------|:-------:|:--------:|:----------:|:---:|:---------:|
> | Generalizable to Reading                       |  -  |         &check;         |    &cross;     |        &check;         |       &check;        |
> | Generalizable to Close-QA                      |   &check; |           -       |    &check;     |         &check;         |       &check;        |
> | Generalizable to Paraphrase                    |  &check;  |         &check;         |    -     |         &check;         |       &check;        |
> | Generalizable to NLI                           |  &check;  |         &check;         |    &cross;     |         -         |       &cross;        |
> | Generalizable to Sentiment                     |  &check;  |         &check;         |    &check;     |         &cross;        |      -        |
>
> (&check; denotes performance better than 0-SHOT)
>
> ## Q7: Presentation improvements: Confusing Figure 2, Prompt Formulation, GPT-Neo as Labeler, Consistent Term.
> Thanks for pointing out these improvements, we will revise the paper accordingly.

---

### Official Review · Reviewer_5iUs · 2023-08-12

**Typos Grammar Style And Presentation Improvements:** The style of Figure 6 can be improved…
**Soundness:** 3

**Excitement:**

3: Ambivalent: It has merits (e.g., it reports state-of-the-art results, the idea is nice), but there are key weaknesses (e.g., it describes incremental work), and it can significantly benefit from another round of revision. However, I won't object to accepting it if my co-reviewers champion it.

**Paper Topic And Main Contributions:**

The paper studies prompt retrieval for zero-shot tasks. Given a task input, the retrieval model tried to select the prompt from a predefined pool that result in highest task success. The retrieval model is trained on clusters of tasks and with a small LM, and is shown to generalize positively to unseen tasks and unseen larger LMs.

While many previous works studied prompt retrieval, this work emphasizes on cross-task and cross-model generalization of the learned retriever. This is achieved by training the retriever on multiple categories of tasks.

**Reasons To Accept:**

- Consistent performance of the learned retriever on multiple large LMs. The paper includes a good coverage of tasks in table 1, and show that the proposed method improves up to 14.6%.
- The method itself is clean and has code included.
- Ablation study compares with fixed/hand-crafted retrievers, illustrating the necessity of learned retriever.
- Good presentation.

**Reasons To Reject:**

- While the paper advertise the prompt retriever to be interpretable, it does not give readers a lot of intuition on what type of prompts are retrieved. A plot visualizing which training task clusters are selected when testing on the held-out cluster would give more intuition about the retriever.

**Reproducibility:**

5: Could easily reproduce the results.

**Reviewer Confidence:**

2: Willing to defend my evaluation, but it is fairly likely that I missed some details, didn't understand some central points, or can't be sure about the novelty of the work.

---

> ### Author Rebuttal · Authors · 2023-08-28
>
> ## Q1: Analysis on the retrieved training clusters for each testing cluster
> Thanks for your suggestion, it provides a very helpful perspective on analyzing the impact of each task type.
>
> ### Q1.1 Which training clusters are retrieved
> We calculate the percentages of retrieved training cluster for each testing cluster, and the table below shows the top-3 training clusters for each testing type:
>
> | Testing cluster | Rank 1 (%) | Rank 2 (%) | Rank 3 (%) |
> |---------------|------------------------|------------------------|------------------------|
> | Reading       | Close-QA (37.5)        | NLI (33.1)             | Summarize (12.5)       |
> | Close-QA      | Reading (80.7)         | NLI (16.9)             | Common Reason (1.1)    |
> | Paraphrase    | Coreference (74.4)     | NLI (13.9)             | Close-QA (6.2)         |
> | NLI           | Paraphrase (47.0)      | Reading (36.1)         | Close-QA (11.5)        |
> | Sentiment     | Summarize (34.9)       | NLI (23.4)             | Reading (17.7)         |
>
> We are unable to provide a visual representation in this rebuttal box, but **the percentage-based visualization will certainly be added** to the next version of our paper.
>
> ### Q1.2: More intuition about the retriever
> Upon analyzing the retrieved clusters, we note that clusters including diverse question types like Reading Comprehension correspond to high retrieved ratios (e.g., 80.7% for Close-QA and 36.1% for NLI), while the less diverse Sentiment Analysis cluster does not reach the top ranks.
>
> Furthermore, we expand this analysis, training two retrievers: UPRISE-read (on Reading Comprehension only) and UPRISE-senti (on Sentiment Analysis only). And conduct cross-task evaluation on the remaining task clusters:
>
> |             |  0-SHOT  |  UPRISE-senti | UPRISE-read |
> |-------------------|--------|-------------|-----------|
> | **Training Data/Prompt Pool** |   **-**  | **Sentiment** | **Reading** |
> | Reading           |   31.6   |     **33.1**      |      -      |
> | Close-QA          |   19.2   |     20.5      |   **24.1**  |
> | Paraphrase        |   47.0   |     **55.3**      |    49.7     |
> | NLI               |   38.3   |     37.0      |   **39.8**     |
> | Sentiment         | 62.7 |       -       |    **63.2**     |
>
> Intuitively and interestingly, a performance trend emerges: UPRISE-senti < UPRISE-read, which echoes cluster retrieved ratios and training data diversity. From this we infer that including tasks of diverse question types in the training data contributes to good generalizability of the retriever.
>
> Inspired by your suggestion, we also figure out the performance decrease on Commonsense Reasoning task type (Q5 for Reviewer zjV3) and acknowledge the impact of prompt format on the task performance.
>
> ## Q2: Improvement of Figure 6
> Thanks, we will add tags/captions to improve Figure 6.

---

### Meta-Review · Area_Chair_hWS1 · 2023-09-19

**Recommendation:** 5

**Metareview:**

The paper proposes a lightweight approach for retrieving prompts from a collection of tasks for an unseen target task.

Pros:

The approach is novel and intuitive

Comprehensive evaluation which demonstrates superior performance in comparison to traditional methods in multiple large LLMs and a wide range of tasks

Additional ablations, analysis of how and why UPRISE works

Cons

Need more intuition about the retriever to back up the claim that the prompts are indeed “interpretable”

More analysis on where and which types of tasks UPRISE does not perform well.

The original work will also benefit from incorporating the additional discussion points/experiments conducted during rebuttal.

---

### Decision · Program_Chairs · 2023-10-07

**Decision:**

Accept-Main

**Comment:**

The paper proposes a lightweight approach for retrieving prompts from a collection of tasks for an unseen target task.

Pros:

The approach is novel and intuitive

Comprehensive evaluation which demonstrates superior performance in comparison to traditional methods in multiple large LLMs and a wide range of tasks

Additional ablations, analysis of how and why UPRISE works

Cons

Need more intuition about the retriever to back up the claim that the prompts are indeed “interpretable”

More analysis on where and which types of tasks UPRISE does not perform well.

The original work will also benefit from incorporating the additional discussion points/experiments conducted during rebuttal.